# Retinal Dystrophies Associated with Mutations in the *RP1* Gene: Genotype–Phenotype Correlations

**DOI:** 10.3390/cimb47030212

**Published:** 2025-03-20

**Authors:** Vito Spagnuolo, Marco Piergentili, Ilaria Passerini, Vittoria Murro, Dario Pasquale Mucciolo, Dario Giorgio, Martina Maccari, Elisabetta Pelo, Ilaria Biagini, Fabrizio Giansanti, Gianni Virgili, Andrea Sodi

**Affiliations:** 1Eye Clinic, Department of Neuromuscular Diseases and Sense Organs, Careggi University Hospital, 50134 Florence, Italy; mpiergent@gmail.com (M.P.); vittoria.murro@unifi.it (V.M.); dario.mucciolo@gmail.com (D.P.M.); dario.giorgio87@gmail.com (D.G.); ilaria.biagini@unifi.it (I.B.); fabrizio.giansanti@unifi.it (F.G.); gianni.virgili@unifi.it (G.V.); 2Department of Neuroscience, Psychology, Pharmacology and Pediatrics (Neurofarba), University of Florence, 50134 Florence, Italy; 3Department of Genetic Diagnostics, Careggi University Hospital, 50134 Florence, Italy; ilariapasserini70@gmail.com (I.P.); peloe@aou-careggi.toscana.it (E.P.); 4Department of Ophthalmology, Livorno Hospital, 57124 Livorno, Italy; 5Eye Clinic, Clinical Trial Center, Careggi University Hospital, 50134 Florence, Italy; martina.maccari@unifi.it; 6IRCCS—Fondazione Bietti, 00184 Rome, Italy

**Keywords:** *RP1* gene, inherited retinal dystrophies, genotype, phenotype, retinitis pigmentosa

## Abstract

Background: We evaluated the genetic and phenotypic features of a cohort of 10 Italian patients affected by Retinitis Pigmentosa (RP) associated with *RP1* sequence variants. Methods: A retrospective, cross-sectional genotype–phenotype correlation study was conducted on a cohort of ten Italian patients (four males and six females) seen at Careggi University Hospital between 2012 and 2024, all affected by RP carrying pathogenic variants in the RP1 gene. A comprehensive ophthalmic assessment and pedigree analysis were performed, focusing on the onset of disease symptoms, the patient’s age at first diagnosis, follow-up duration, and the presence of comorbidities. Results: Our cohort included ten Italian patients with a mean age of 59 (range of 32–79 years). The median age when symptoms first presented was 43 years (range of 2–74), with a mean follow-up period of 9.3 ± 2.6 years. The main symptoms at presentation were hemeralopia and visual field constriction. Fundus examination revealed a classic RP phenotype. Fundus autofluorescence (FAF), optical coherence tomography (OCT), Electroretinogram (ERG), and visual field testing confirmed the typical features of classic retinitis pigmentosa in most cases. Conclusions: This single-center cohort of Italian patients provides insights into the clinical and genetic characteristics of *RP1*-associated RP. By comprehensively identifying genetic variations and their associated clinical manifestations, therapeutic interventions targeting specific genetic abnormalities can be better tailored. This approach holds promise for improving the prognosis and quality of life for individuals with *RP1*-associated RP.

## 1. Introduction

Inherited retinal dystrophies (IRDs) represent a clinically and genetically heterogeneous spectrum of conditions that affect photoreceptor cells, leading to visual impairment [1,2]. Among retinal dystrophies, retinitis pigmentosa (RP) (OMIM # 268,000) predominantly affects rod photoreceptors and has the highest worldwide prevalence of 1 in 4000 individuals. The condition primarily follows three patterns of inheritance: autosomal recessive (ARRP), autosomal dominant (ADRP), and X-linked RP (XLRP). Up to 50% of cases are isolated (simple or sporadic RP, SRP) with no family history [3].

The onset, progression, retinal manifestations, and visual prognosis vary considerably among RP cases. In the early stages, rod degeneration results in night blindness, the loss of peripheral vision, and tunnel vision. In later stages, as cone photoreceptors become affected, patients may experience dyschromatopsia and progressive central vision loss [4].

IRDs often present as isolated findings without additional systemic manifestations (non-syndromic IRD), with *RP1* gene variants identified as potential causative factors. However, in some cases, retinal degeneration is part of a broader systemic disorder (syndromic IRD) [5].

The *RP1* gene is localized to the pericentric region of chromosome 8 (locus 8q11.23-q12.1) and encodes a 2156-amino-acid protein primarily located in the photoreceptor connecting cilia and axoneme. The N-terminal segment includes a microtubule-binding domain (DCX), which is essential for photoreceptor longevity because it stabilizes microtubules and regulates cilia length [6]. The *RP1* mRNA is exclusively expressed in the photoreceptor cell bodies and inner segments of the retinas [7]. The *RP1* protein regulates the stability and length of the microtubule-based axoneme in photoreceptors, playing a critical role in the differentiation of photoreceptor cells. It ensures the correct organization of the outer segment of rod and cone photoreceptors, maintaining the proper orientation and stacking of outer segment disks along the photoreceptor axoneme.

*RP1* mutations are associated with autosomal dominant retinitis pigmentosa [8,9,10] but can also cause autosomal recessive forms of the disease [11,12]. To date, at least 170 *RP1* mutations have been reported, accounting for approximately 5–10% of autosomal dominant retinitis pigmentosa (ADRP) cases and about 4.5% of autosomal recessive retinitis pigmentosa (ARRP) cases. However, prevalence varies among studies and populations [13,14,15]. Recently, D’Esposito et al. reported a high prevalence of the *RP1* p.Ser740 variant in ADRP* within a specific Italian cohort, suggesting the presence of a founder effect [16]. Furthermore, *RP1* sequence variants have also been associated with other phenotypes, such as cone–rod dystrophy and macular degeneration [17,18].

The c.2029C>T (p.Arg677*) nonsense variant is the most frequently reported dominant pathogenic variant in *RP1* and was also the most prevalent in our cohort [19,20]. Most *RP1* gene mutations are single-nucleotide substitutions that create a premature stop codon or insertions/deletions, leading to a truncated protein [19,21].

This study investigates the genetic and phenotypic characteristics of a cohort of Italian patients with RP carrying *RP1* sequence variants. Previous studies reported on some patients of different ethnic groups with a clinical diagnosis of RP associated with *RP1* mutations [9,10]. In this paper, we specifically focus on an Italian group of *RP1*-associated RP, trying to evaluate possible genotype–phenotype correlations. Italian *RP1* RP patients have already been included in previous large genetic epidemiologic studies [13], mainly those reporting on the molecular genetic results and not specifically considering genotype–phenotype correlations.

## 2. Materials and Methods

This retrospective, cross-sectional genotype–phenotype correlation study, conducted at Careggi Hospital in Florence, adhered to the principles outlined in the Declaration of Helsinki. A retrospective chart review of clinical notes of patients referred to our Retinal Degeneration Referring Center between 2012 and 2024 was conducted. Only patients with a confirmed genetic diagnosis of *RP1*-related RP were enrolled. Ten patients (four males and six females) carrying pathogenic variants in the *RP1* gene were included in this study.

A comprehensive ophthalmic examination included assessments of best-corrected visual acuity (BCVA), refraction, intraocular pressure (IOP) measurements, and anterior and posterior segment biomicroscopy. Detailed clinical histories were collected for all patients, with a particular focus on the onset of disease symptoms, the patient’s age at first diagnosis, follow-up duration, the possible presence of extraocular associations, and any family history of the condition.

Imaging modalities included wide-field ocular fundus photography (Optos PLC, Dunfermline, UK), spectral-domain optical coherence tomography (OCT; Heidelberg Engineering, Heidelberg, Germany; Carl Zeiss Meditec, Jena, Germany), and Goldmann visual field testing. Electroretinography (ERG) was performed according to the standard protocol of the International Society for Clinical Electrophysiology of Vision (ISCEV) [22]. After pupil dilation, ERG recordings were obtained using the Retiscan 201 B4 system (Roland Consult, Brandenburg, Germany). Results were compared with those of an age-matched control group (n = 15) with no significant ocular pathology who attended our clinic. All data were recorded in an Excel database (Microsoft Excel 2010, Microsoft Office Professional Plus 2010).

### Molecular Genetic Analysis

Following informed consent and a pre-test genetic counseling, 10 mL of peripheral blood was obtained from the antecubital vein using EDTA-containing vials. DNA was extracted from 200 μL of peripheral blood by using automated DNA extractors, BioRobot EZ1 (QIAGEN GmbH, Hilden, Germany) or QIAsymphony SP workstation (QIAGEN GmbH, Germany), according to the manufacturer protocols. For each family, the proband was the first patient with a clinical diagnosis of RP included in the study. In 10 patients from nine independent pedigrees, genetic analysis was performed with targeted next-generation sequencing (NGS) at the Department of Genetic Diagnosis (Careggi Teaching Hospital, Florence, Italy) which is a certified UNI EN ISO 9001:2008 laboratory. A panel of 137 genes known to be associated with retinal dystrophies was used for targeted NGS; Exons of DNA samples were captured and investigated as shown previously with enrichment methodology SureSelect QXT Target Enrichment (Agilent Technologies, Santa Clara, CA, USA), using the Illumina NextSeq TM500 platform (Illumina, San Diego, CA, USA). All identified variants were confirmed with Sanger sequencing and further segregated into the respective families when other relatives were available. Variations were annotated using Alissa Interpret Rev. 5.4.2 Agilent Inc. (Santa Clara, CA, USA) by comparing with other databases (1000 Genome project, Exome aggregation consortium, Ensmble, dbSNP, ClinVar, Human Gene Mutation Database (HGMD), Variation Database (LOVD), RetinoGenetics, RetNet, Mutation Database of Retina International) and were predicted for pathogenicity with online bioinformatic tools (REVEL, SIFT, PolyPhen, Mutation taster, Mutation assessor, and Variant Effect Predictor). We evaluated allele frequencies (GnomAD), a co-segregation analysis, and published data. The variants were classified according to the current revised guidelines of ACMG [23].

## 3. Results

### 3.1. Phenotype–Genotype Correlation in RP1-Associated RP

Our cohort included ten Italian patients with a median age of 59 (range of 32–79 years) affected by *RP1*-associated RP. No extraocular signs indicative of syndromic RP were present. The patients’ demographics and clinical data are summarized in Table 1. While P8 and P9 are sisters, no other patients were related in this study. Visual acuity in our population showed high variability, with more severe phenotypes demonstrating visual acuity ranging from hand motion (HM) to light perception (LP), while milder phenotypes reached values close to normal.

The median age at which symptoms first appeared was 42 (range of 2–74 years). As expected, patients with an earlier onset of the disease (e.g., P2, P3, P7, and P10) tended to show a worse prognosis. However, a poor prognosis was also observed in some patients presenting with later-onset symptoms (e.g., P5 and P8). The mean follow-up period was 9.3 ± 2.6 years (range of 6–12 years). The most common symptom at onset was hemeralopia (50%), followed by visual field constriction (40%). One patient (P3) presented with nystagmus as the initial feature. Interestingly, as nystagmus in children can be associated with various ocular or neurological conditions [24], this patient received an early diagnosis at the age of 2 years. On fundus examination, most patients exhibited the classical features of RP, including diffuse retinal dystrophy with bone spicules and dark clumps of pigment in the midperiphery, attenuated retinal vessels, and optic disc pallor. Among the ten patients (twenty eyes), eight eyes were pseudophakic, six showed predominantly posterior cortical cataracts, and six had clear lenses.

Imaging with FAF revealed hypo-autofluorescence corresponding to areas of RPE atrophy. In one patient (P6), a perimacular hyper-autofluorescent ring was observed at the border between the atrophic and spared macular retina (Figure 1).

OCT images revealed abnormalities in the outer retina including a disrupted ellipsoid zone (EZ) and thinning of the outer nuclear layer). Complications such as a cystoid macular edema (CMO) (Figure 2) and an epiretinal membrane were observed in five (25%) and two (10%) eyes. More advanced disease was noted in three eyes, which developed macular atrophy and resulted in poor vision.

ERG testing revealed reduced to absent photopic and scotopic responses in all patients. Goldmann perimetry consistently demonstrated progressive concentric constriction and, in some cases, a tubular visual field. The visual field was bilaterally not evaluable in five patients with very poor visual acuity (P2, P5, P7, P8, and P10).

### 3.2. Genetic Characteristics

The sequence variants identified in our series are summarized in Table 1. All the mutations have already been reported in the literature [11,12,13,25,26].

The most prevalent mutation was the c.2029C>T (p.Arg677*) variant, which has been previously reported as moderately common in other studies [21]. This mutation leads to the production of a truncated protein lacking 50–70% of the C-terminal. It has been identified in various populations (European, American, Asian, and African) and is a recurrent mutation among European-descendant RP patients, accounting for approximately 3% of dominant RP cases in North America [7].

In our series of nine independent pedigrees (with patients P8 and P9 being sisters), seven patients carried a heterozygous *RP1* mutation with autosomal dominant inheritance, while the remaining three patients had *RP1* biallelic mutations (homozygous duplications) and pedigrees consistent with a recessive mode of inheritance (P2, P7, and P10).

## 4. Discussion

In this study, we investigated the genetic and phenotypic profiles of a cohort of ten Italian retinitis pigmentosa (RP) patients carrying *RP1* mutations. On fundus examination, most patients exhibited a typical RP phenotype, as illustrated in Figure 3.

Clinically, we observed the coexistence of mild phenotypes alongside more severe clinical presentations with extensive impairment of visual function. In our study, patients with an early onset of the disease and longer symptom duration, as expected, exhibited a severe clinical phenotype (P2, P7, and P10, with onset at 4, 12, and 6 years of age, respectively). However, a poor prognosis was also observed in some patients presenting with later-onset symptoms (P5, P8). Interestingly, patients with the same pathogenic variant in the *RP1* gene, such as P1 and P5, who had a relatively similar onset of symptoms, showed markedly different visual prognoses. This variability could potentially be explained by the effect of genetic modifiers, which may have contributed to variable clinical outcomes, as well as the influence of unknown environmental factors. Our data confirm the possible association of *RP1* sequence variants with both autosomal dominant and recessive modes of disease transmission. However, we did not observe significant clinical differences between the dominant and recessive forms of the disease. Both groups in our study predominantly presented with a classic RP phenotype, encompassing both mild and more severe clinical presentations. Among patients with bilateral severe vision loss (visual acuity bilaterally reduced to HM or LP), two carried a heterozygous mutation (P5 and P8), while the other three (P2, P7, and P10) were homozygous for the c.5962dup *RP1* variant. This genotype was previously reported in Spanish patients affected by RP [26]; however, clinical details were not available for comparison. Autosomal recessive forms of RP associated with *RP1* mutations are reported to exhibit significantly more severe disease progression than autosomal dominant cases [12]. However, this was not observed in our study. The limitations of this study, including the small number of enrolled patients, prevent a clear evaluation of the clinical differences between *RP1*-dominant and -recessive RP phenotypes. Nevertheless, our findings demonstrate that similar clinical scenarios are possible across inheritance patterns. As previously reported [7,27,28], the c.2029C>T (p.Arg677*) was the most prevalent *RP1* mutation in our series. Interestingly, this mutation was not associated with a specific geographic area of origin, as the patients came from different Italian regions (Lombardia, Tuscany, Marche, and Sicily).

## 5. Conclusions

In conclusion, we reported the clinical and genetic data of a cohort of Italian patients with a clinical diagnosis of RP associated with mutations in the *RP1* gene. Our results confirm the possible association of *RP1* sequence variants with both dominant and recessive modes of inheritance and further expand the genetic and phenotypic characterization of a specific Italian population. We also observed a high degree of variability in both the onset of the disease and its prognosis, emphasizing the importance of identifying new prognostic factors involved in disease progression. Furthermore, a better understanding of the onset and progression of the disease will be crucial for delivering future treatments in an effective and timely manner.

## Figures and Tables

**Figure 1 cimb-47-00212-f001:**
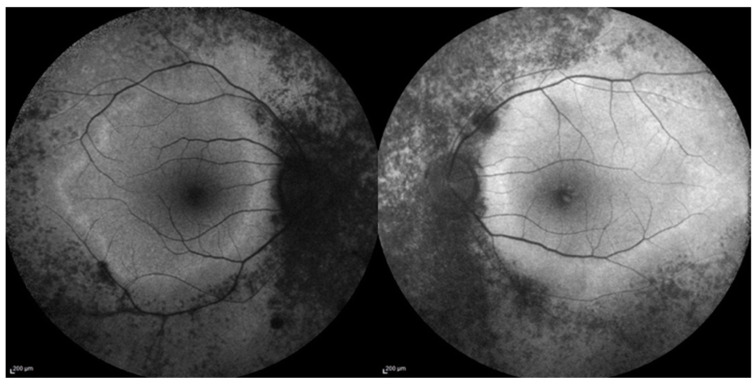
Fundus Autofluorescence (FAF) of the right and left eye of 52-year-old male patient (P6), show symmetric areas of mottled hypo-autofluorescence corresponding to the dystrophic retina, starting from the optic disc. There is also a typical ring of hyper-autofluorescence delimiting the spared retina.

**Figure 2 cimb-47-00212-f002:**
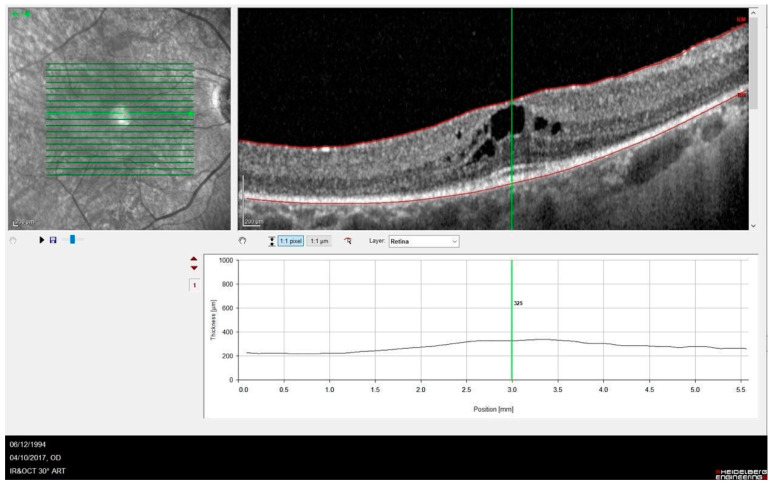
SD-OCT showing a case of RP complicated by cystic macular edema (P1, 66-year-old female patient).

**Figure 3 cimb-47-00212-f003:**
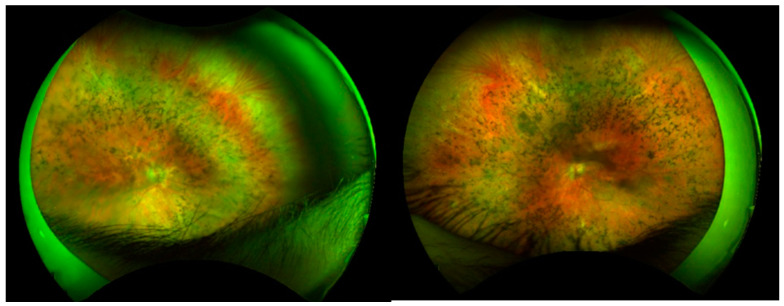
Fundus appearance of P7 (male, 32 years old) with a typical RP pattern.

**Table 1 cimb-47-00212-t001:** Patients’ demographics and clinical data.

Patient	Gender	Age	Symptoms at Time of Diagnosis	Age of Symptom Onset	BCVA logMAR (RE; LE)	Genetic Variants
1	F	66	Reduced visual field	55	0.1; 0.1	*c.2029C>T p.(Arg677*)/WT*
2	F	58	Hemeralopia	4	HM; HM	*c.5962dup p.(Ile1988Asnfs*3)/c.5962dup p.(Ile1988Asnfs*3)*
3	F	32	Nystagmus	2	HM; 0.7	*c.2019C>T p.Arg677*)/WT*
4	M	60	Hemeralopia	43	0.2; 0.5	*c.2019C>T p.Arg677*)/WT*
5	F	77	Hemeralopia	47	LP; LP	*c.2029C>T p.(Arg677*)/WT*
6	M	52	Reduced visual field	48	0.1; 0.1	*c.2029C>T p.(Arg677*)/WT*
7	M	32	Hemeralopia	12	HM; HM	*c.1234dup (p.Met412Asnfs*7)/c.1234dup (p.Met412Asnfs*7)*
8	F	79	Reduced visual field	74	LP; no LP	*c.2029C>T p.(Arg677*)/WT*
9	F	77	Hemeralopia	30	0.8; 1.0	*c.2029C>T p.(Arg677*)/WT*
10	M	57	Reduced visual field	6	LP; LP	*c.5962dup p.(Ile1988Asnfs*3)/c.5962dup p.(Ile1988Asnfs*3)*

## Data Availability

The original contributions presented in the study are included in the article, further inquiries can be directed to the corresponding authors.

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
