# Peer review of "Retinal Dystrophies Associated with Mutations in the *RP1* Gene: Genotype–Phenotype Correlations"

_cimb, 2025, doi:10.3390/cimb47030212_

Round 1

Reviewer 1 Report (Previous Reviewer 1)

Comments and Suggestions for Authors

Dear Authors, 

Thank you for reviewing and sending this new version of your article "Retinal dystrophies associated with mutations in the RP1 gene: 2 genotype-phenotype correlations"

I think this version is improved, although there are a few parts requiring further editing.

For example:

Row 524 "However, poor prognosis was observed also in some patients presenting with symptoms in older age (P5, P8)."I think later onset would be more appropriate than older age". 

I think the Discussion section needs some adjustments:

for example, row 525: "patients with the same genetics as P1 and P5" could be replaced with "patients with the same genetic variant as P1 and P5" to be more accurate.  In practice, we do not really use the term 'genetics' to refer to genetic variants.

In fact, phenotype variability is well documented in most genetic disorders, including in people belonging to the same family.  I think this is reflected by the small cohort of RP patients presented here and it is not really a new element.

I think this sentence can be re-written to improve clarity: rows 585 to 587: "Among patients with bilateral severe vision loss (visual acuity bilaterally reduced to HM or LP), two carried a heterozygous mutation (P5, P8) while the other three (P2, P7 and P10) carried a biallelic mutation, which resulted the c.5962dup/c.5962dup variant in two of those." Instead of saying "c.5962dup/c.5962dup variant" you could say that 2 patients were homozygous for the c.5962dup RP variant. 

Author Response

Thank you for helping us improving our paper. We have corrected as suggested all the parts requiring editing.

Reviewer 2 Report (New Reviewer)

Comments and Suggestions for Authors

The article reports the findings from a retrospective cases series of patients with retinitis pigmentosa carrying pathogenic variants of the RP1 gene. The study was aimed at elucidating the genotype-phenotype correlation.

The article is well structured and adequately concise.

However, there are several limitations. The authors may address some issues to improve overall quality of the manuscript, though the intrinsic limit is linked to the study protocol and to the cohort size. A few remarks are reported.

Methods:

The study cohort comprises 10 patients carrying RP1 variants. How were they enrolled – i.e. from what cohort of eligible individuals, in what period?

Results:

Were all cases affected with isolated RP? Were there other, extra-ocular signs?

Inherited retinal dystrophies are inherited diseases. The article failed to mention the family history. Were all sporadic cases? Anyway, family history should be detailed for all cases. As family history was not reported, it was not described whether the RP1 variants co-segregate with the disease, though the methods mentioned that segregation analysis was carried out. Moreover, it was not specified whether the carriers of biallelic variants were confirmed to be compound heterozygous – i.e. that the variants are in trans – and whether the heterozygous variants were paternally or maternally transmitted, or were de novo.

The methods section reports that a panel of IRD-related genes was investigated. However, the findings in the RP1 gene were reported. Were there pathogenic variants and variants of unknown significance in other genes? Given the number of assessed genes, several VUS may have occurred.

The classification according to the ACMG guidelines should be reported for all RP1 variants found.

Table 2 should be merged with table 1 to improve readability of data.

Discussion:

The authors found heterogeneous clinical manifestations as associated with the same RP1 genotype – that is not astonishing in IRD. The discussion, nevertheless, deserved to be better addressed and referenced [lines 212-214]. Besides, the term ‘patients with the same genetics’ is rather colloquial. As well, the 3.2 title ‘Genetics’ should be more specific.

The conclusion ‘These findings may contribute to the development of future therapeutic trials…’ is an overstatement.

Comments on the Quality of English Language

Language remarks:

The authors used the term ‘prognosis’ when the clinical course was presumably intended – prognosis rather relates with the prediction of the clinical course.

The term ‘Caucasian’ is a non-sense in the context of genetic studies. Did the authors mean ‘of European descent’?

All gene names should be in italics throughout the text.

A few typos should be corrected.

See also the last remark above.

Author Response

Thank you for helping improving the quality of our article. 

The text has been modified accordingly in the parts requiring editing. We although were unable to provide a complete family history for all members as some informations were missing. We included the relevant informations in the main text. 

Both tables have been merged and mutations typed in italics. 

No syndromic cases were found in our cohort. 

Round 2

Reviewer 2 Report (New Reviewer)

Comments and Suggestions for Authors

The previous the peer-review process had raised some relevant issues, as related to the study findings and to the interpretation of results. The remarks deserved point-by-point response arguments. Conversely, the authors’ response is rather vague – and somewhat inconsistent [e.g.: the authors got mixed with the nomenclature and confused ‘gene name’ with ‘gene variant’ – see the reviewer’s suggestion].

Moreover, in the current version for peer review the last revision was stacked to a previous one; a substantial part of the manuscript appears as revised, thus hampering to assess the actual changes and whether the revision accounted for the reviewer’s remarks. Some relevant sections of the manuscript apparently changed, with no justification in the authors’ note.

Comments on the Quality of English Language

The English language still needs to be checked throughout the manuscript.

Author Response

Thank you for helping improving the quality of our article. 

The text has been modified accordingly in the parts requiring editing.

This manuscript is a resubmission of an earlier submission. The following is a list of the peer review reports and author responses from that submission.

Round 1

Reviewer 1 Report

Comments and Suggestions for Authors

Dear Authors, 

It was a pleasure to review your brief report describing clinical and genetic findings in people with retinal dystrophies and variants in the RP1 gene.

I thought it is very useful to have a set of additional cases reported for this rare disorder.

1) in your introduction mentioned the world wide prevalence of retinitis pigmentosa (RP), which is quite common for a rare condition.  However, since RP is so heterogeneous from the genetic point of view, I think it would be relevant for your article to add the proportion of cases with an RP1 variant, if possible, by comparing your cohort with other cohorts reported previously.  Together with the clinical information, this would give an idea on a likely diagnostic yield and guide genetic testing in those regions where large panels are not available.

2) The pathogenicity assessment of the missense genetic variant was based on a few by informatics tools but it would be useful to add Revel which is currently used in clinical diagnostic labs.

3) I suggest replacing the term 'mutation' with the term 'variant' according to the current recommendations for publications reporting genetic results.

4) Table 2 showing the genetic variants detected in these 12 patients would benefit from an additional column to indicate whether these variants were homozygous or heterozygous and whether they were de novo.

5) in the section dedicated to the genetic results obtained for these 12 patients, there are a few errors:

5a) A new variant c.5753G>A (p.Cys1918Tyr) is mentioned in the text (Row 160) but I can see that this is not included in table 2 and I wonder whether this is an error.  This variant can be found reported as of unknown significance in ClinVar under the accession number: RCV003889499.1.  If this is indeed a variant found in 1 of the patients included in this study, I think it would be relevant to reclassify it and consider whether there are sufficient elements to confirm classification as likely pathogenic.

5b) the variant for patients 3 in table 2 is inaccurately described: c.118C>T is not correct and cannot result in the replacement of threonine 373 as 118 would involve a variant affecting codon 39, not 373.  It is possible that the name of this variant misses a figure, as c.1118C.T would indeed result in p.Thr373Ile.  This variant has also been reported in ClinVar (VCV000005970.34) and it is currently classified as benign/likely benign.  This means that if your patient carry this variant, this would not be clinically significant and needs to be removed.  Interpretation of missense variants in genes where loss of function genetic variants are confirmed causes of disease, needs additional precaution, before deciding to guide clinical management.

6) I thought the reference list can be supplemented, for example with this article:  Riera M, AbadMorales V, Navarro R, et al. Br J Ophthalmol 2020;104:173–181 

There may be other case studies, as well as more recent ClinVar database entries that can help better define the possible genetic cause is in your cohorts.

Comments on the Quality of English Language

Only minor/moderate editing needed. 

Reviewer 2 Report

Comments and Suggestions for Authors

Review of Spagnuolo V. et al., “Retinal dystrophies associated with mutations in the RP1 gene: Genotype-phenotype correlations.”

The study conducted by Spagnuolo V. et al. presents a cohort of 12 Italian patients with Retinitis Pigmentosa associated with RP1 mutations. While the study contributes valuable data to the field, it lacks sufficient analytical rigor in dissecting the depth of the findings. The figures presented appear more illustrative rather than analytical, which diminishes their scientific impact. If the standard deviation is very high, the values should be reported as a range for easy interpretation (e.g. disease onset 2-72 Years of age).

The authors should aim to create figures that present meaningful scientific arguments. For example, fundus images could be categorized according to disease severity, such as severe or less severe cases or any other meaningful analytical criteria, supported by appropriate scientific arguments. Furthermore, any challenges faced during diagnosis could be highlighted to provide a comprehensive view of the study or the authors should highlight the important new scientific information on RP1-associated retinal dystrophy resulting from this study.

Moreover, the figures lack patient details such as age, sex, and proper annotation, that what the author wants readers to show. Including these information and proper annotation to the figures or making more tables that provides more details would enhance the interpretability and value of the data presented. All figures should be appropriately cited and discussed within the main text of the results section.

The authors must incorporate the above mentioned suggestion and revise  the manuscript in a way so readers could benefits including clinicians and other researchers.

Comments on the Quality of English Language

English is okay.

Reviewer 3 Report

Comments and Suggestions for Authors

The manuscript “Retinal dystrophies associated with mutations in the RP1 gene. Genotype-phenotype correlations” describes a cohort of RP patients with genetic variants in the RP1 gene. While not a novel idea, the manuscript provides relevant information. However, I feel there is a lot of information lacking and the ideas are quite disorganized and not supported by evidence from the results. Here are my suggestions:

·     Please use all caps in the abbreviations, specifically, ARRP, ADRP and XLRP.

·     I think the Introduction section should have less paragraphs (for example, merging line 49 with 50, and 54 with 55) as this enhances reading fluidity.

·     Overall, the contents of the Introduction provide an adequate summary of the background information required for this work.

·     The Methods section are appropriately described; however, the paragraph starting on line 86 has repeated information from the previous (i.e. the listing of imaging modalities). I would suggest including the specifics of each imaging modality in between parenthesis after it is firstly mentioned.

·     It is mentioned that P10 and P11 patients are family members – please state that no other patients are related in the beginning of the Results section.

·     I would suggest to expand Table 1 in order to include additional columns that provide information regarding findings on the different imaging tests performed (OCT, ERG…), as the textual description is not objective (“Most of the patients…”, “…in some patients…”).

·     The Genetics section is quite incomplete/incoherent, and requires significant improvement:

o  First, it is not stated which patients have biallelic variants. Furthermore, authors mention two patients are compound heterozygous, yet none of the patients have two genetic variants listed.

o  Then, no classification is provided. Authors state the usage of ACMG criteria, but do not add the final classification and the selected criteria. They also state usage of a “dedicated software” which I strongly suggest not to use, as automatic classification systems are well-known not to provide accurate information most of the times.

o  Table 2 requires some formatting and consistency review.

o  Are there any genotype-phenotype correlations? This is indicated in the title of the paper (which should be corrected, as there should not be any full stop in titles), but authors never assess it in the results section, only mentioning it in the discussion. Objective evidence regarding this should be provided.

I hope the authors consider this criticism as constructive and hope it provides useful for the improvement of their work.

Comments on the Quality of English Language

Moderate revision of English is required, mostly related with syntax.
